# Knee Diameter and Cross-Sectional Area as Biomarkers for Cartilage Knee Degeneration on Magnetic Resonance Images

**DOI:** 10.3390/medicina59010027

**Published:** 2022-12-23

**Authors:** Elias Primetis, Dionysios Drakopoulos, Dominik Sieron, Hugo Meusburger, Karol Szyluk, Paweł Niemiec, Verena C. Obmann, Alan A. Peters, Adrian T. Huber, Lukas Ebner, Georgios Delimpasis, Andreas Christe

**Affiliations:** 1Department of Radiology SLS, Inselgroup, Bern University Hospital, University of Bern, Freiburgstrasse 10, 3010 Bern, Switzerland; 2Department of Physiotherapy, Faculty of Health Sciences in Katowice, Medical University of Silesia in Katowice, 40-752 Katowice, Poland; 3District Hospital of Orthopaedics and Trauma Surgery, Bytomska 62 St., 41-940 Piekary Slaskie, Poland; 4Department of Biochemistry and Medical Genetics, Faculty of Health Sciences in Katowice, Medical University of Silesia in Katowice, 40-752 Katowice, Poland; 5Department of Diagnostic, Interventional and Pediatric Radiology, Inselspital, Bern University Hospital, University of Bern, 3010 Bern, Switzerland

**Keywords:** Outerbridge, chondromalacia, aging, body mass index, degeneration, magnetic resonance imaging

## Abstract

*Background and Objectives*: Osteoarthritis (OA) of the knee is a degenerative disorder characterized by damage to the joint cartilage, pain, swelling, and walking disability. The purpose of this study was to assess whether demographic and radiologic parameters (knee diameters and knee cross-sectional area from magnetic resonance (MR) images) could be used as surrogate biomarkers for the prediction of OA. *Materials and Methods*: The knee diameters and cross-sectional areas of 481 patients were measured on knee MR images, and the corresponding demographic parameters were extracted from the patients’ clinical records. The images were graded based on the modified Outerbridge arthroscopic classification that was used as ground truth. Receiver-operating characteristic (ROC) analysis was performed on the collected data. *Results*: ROC analysis established that age was the most accurate predictor of severe knee cartilage degeneration (corresponding to Outerbridge grades 3 and 4) with an area under the curve (AUC) of the specificity–sensitivity plot of 0.865 ± 0.02. An age over 41 years was associated with a sensitivity and specificity for severe degeneration of 82.8% (CI: 77.5–87.3%), and 76.4% (CI: 70.4–81.6%), respectively. The second-best degeneration predictor was the normalized knee cross-sectional area, with an AUC of 0.767 ± 0.04), followed by BMI (AUC = 0.739 ± 0.02), and normalized knee maximal diameter (AUC = 0.724 ± 0.05), meaning that knee degeneration increases with increasing knee diameter. *Conclusions*: Age is the best predictor of knee damage progression in OA and can be used as surrogate marker for knee degeneration. Knee diameters and cross-sectional area also correlate with the extent of cartilage lesions. Though less-accurate predictors of damage progression than age, they have predictive value and are therefore easily available surrogate markers of OA that can be used also by general practitioners and orthopedic surgeons.

## 1. Introduction

Osteoarthritis (OA) of the knee is a very common pathology affecting the whole joint but in particular the cartilage [1]. It is classified as primary if its origin is unknown, or secondary if it is subsequent to a specific condition or event like trauma, repetitive micro stress, surgery, or malalignment [2]. OA is a degenerative disease that usually becomes manifest in the elderly [1]. The medial compartment is the most affected, though also the lateral and the patellofemoral compartments are also affected. The cartilage changes characteristic of this condition are accompanied by marginal osteophytes, subchondral bone cyst formation (geodes) and sclerosis, buttressing, and soft tissue changes, namely ganglion cyst formation, and periarticular soft tissue edema. Reactive synovium thickening and intra-articular fluid can also be observed [3,4]. The symptoms, including pain, stiffness and walking disability can increase to the point to deteriorate life quality, leading in some cases to depression [5]. The pathology is prevalent in the obese female population, and correlates with knee diameter [6,7,8,9]. Disease frequency and severity increase with increasing age [10,11].

Obesity, previous knee trauma, biomechanical factors, female gender and older age are common known risk factors for the development of osteoarthritis [12,13,14]. As no drugs are available to treat this condition, the focus is on prevention and management of joint damage progression through interventions on the modifiable risk factors, like the diet and physical exercise [13].

MR imaging (MRI) with 1.5 and 3 Tesla systems has proven to be a suitable technology for both the quantitative and qualitative assessment of joint cartilages [15,16], especially of the knee, and is clearly superior to plain radiography for the evaluation of the hyaline cartilage, as the last methodology provides only indirect signs of chondromalacia, actually not depicting the cartilage itself [17,18,19,20]. The Outerbridge scoring system is widely used to evaluate the extent of cartilage degeneration. It is a 5-grade scale: grade 0—normal articular cartilage, grade 1—softening of the cartilage due to biochemical modifications of the cartilage composition, grade 2—extent of cartilage loss <50%, grade 3—extent of cartilage loss >50%, and grade 4—complete cartilage loss accompanied by subchondral bone changes [21].

In spite of the high accuracy of MRI techniques for the assessment of the disease extent, this methodology is expensive and its use is restricted to equipped radiologic facilities and experienced medical staff. Therefore, more easily available tools to detect the disease and monitor its progression are needed.

Starting from the evidence that obese individuals with large knee cross sections often present with extensive cartilage degeneration (chondromalacia) [6], we hypothesized that both knee diameter and cross-sectional area might be employed as easily accessible surrogate biomarkers of chondromalacia. Similarly, given the high prevalence of OA in the obese female population as well as in the elderly [1,7], further parameters like age, gender, and BMI might also be suitable predictors of OA. 

## 2. Materials and Methods

### 2.1. Ethics

The responsible ethics commission waived the requirement to obtain informed consent from the patients included in this study due to the study’s retrospective nature and the irreversible anonymization of patients’ identifying information.

### 2.2. Study Design

Among patients undergoing knee MRI between 2018 and 2019, 481 patients were retrospectively selected from the archives of community and clinical hospitals as well as private clinics in Zamość Elblag, Jelenia Góra and Bielsko-Biala (Poland) based on the inclusion and exclusion criteria listed below. The study was performed according to the STROBE guidelines. A number of 120–121 patients for each of four age groups were selected: <30 years (120); 30–45 years (120); 46–60 years (120); >60 years (121). Demographic data were retrieved from the MRI safety questionnaires of the corresponding clinics (age, sex, height, weight and BMI). The MR images used for this study had been acquired on either 1.5 T (SIGNA, GE, Milwaukee, WI, USA) or 3 T (Ingenia 3.0 T, Philips, Amsterdam, Netherlands) scanners using the following diagnostic sequence protocols: axial, sagittal and coronal PD FS, sagittal and coronal T1 (all with a slice thickness of 3 mm), 3D high-resolution PD FS with a slice thickness from 0.8 to 1 mm. All MRI data were irreversibly anonymized, and evaluated using iMac pro (Apple, Cupertino, CA, USA) with FDA-approved OsiriX MD software (version 11.0, Pixmeo SARL, Bernex, Switzerland). The radiological evaluation of cartilage chondromalacia included the medial, lateral and retropatellar compartments of the knee joint. 

### 2.3. Inclusion and Exclusion Criteria

All patients undergoing an MRI of the knee in the period from 2018 to 2019 at the above-mentioned radiological clinics were eligible for this study. MRI referral was mostly due to knee pain, suspicion of arthrosis or post-traumatic lesions. 

Patients with previous surgery, chronic post-traumatic changes, tumorous lesions in the knee joint, and examinations lacking the PD sequences for any reason (abortion of the MRI scan) were not included in the study.

### 2.4. Image Analysis

Four radiologists with radiology board examination and 10, 12, 17 and 25 years of experience in musculoskeletal radiology analyzed the images. Each radiologist assessed 30 patients from each age group. No double reading was performed, thus inter- and intrareader agreement could not be assessed. To evaluate cartilage chondromalacia, the modified Outerbridge classification (Table 1) for arthroscopic cartilage evaluation was used, and grading scores were attributed based on fat-saturated proton density sequences [22,23]. Besides cartilage assessment using Outerbridge scoring system, the Insall-Salvati index, cross-sectional area of the knee at the level of the upper pole of the patella as well as diameters at the same level were calculated [24]. 

The following parameters of the knee joint were measured for the subsequent analysis:
(1)On the sagittal PD-weighted images, the patellar ligament and the max. pole to pole distance in the patella were measured to calculate the Insall-Salvati index (Figure 1).(2)Knee diameters were measured on axial PD-weighted images at the exact level of the patella upper pole (strictly antero-posterior = vertical, and medio-lateral = horizontal, independent of the leg position; Figure 2). The larger of these 2 diameters was taken as “maximal diameter”.(3)The whole knee cross-sectional area at the same level was automatically calculated by the Osirix software (Figure 2),(4)The largest axial diameter of the distal femur at the level of maximal condyle diameter was measured and used for normalization (Figure 3).

In addition, some individuals tend to use the right more than the left side, or vice versa, and/or may be or may have been active in a sport discipline involving overuse of one side and thus are suffering from repetitive microtrauma. This is potentially related with a differential Outerbridge grading for the two body sides, and therefore the analysis did include specification of the investigated side.

Furthermore, the MRI resolution of images acquired with a 3 T unit is higher than that of images from examinations performed at a 1.5 T unit, suggesting the possibility of earlier detection of (severe) degeneration related with the use of the more powerful 3 T unit [19]. Therefore, the used MRI strength (1.5 T vs. 3 T) Tesla was also included in the evaluation. 

### 2.5. Statistical Analysis

The Outerbridge scores were assessed in the lateral, medial and retropatellar compartment. The maximal score of all three compartments was pooled into zero or slight degeneration (Outerbridge 0–2) and severe degeneration (Outerbridge 3 and 4) and was defined as the classifying variable (outcome). A logistic regression was performed with the binary variables (gender (f, m), examination side (right, left), MRI strength (1.5 or 3 T)) to retrieve regression coefficients and odds ratios. Measurements were normalized for 122 patients: the maximal knee diameter and cross-sectional area were normalized by dividing their value by the max femoral diameter to level out the fact that larger patients have larger knees without necessarily suffering from knee degeneration. The non-binary variables (age, height, weight, BMI, Insall-Salvati index, knee cross-sectional area, vertical, horizontal and maximal diameter as well as normalized maximal diameter and cross-sectional area) were analyzed using receiver operating characteristic curves (ROC). The respective areas under the curve (AUC) were used as comparators to establish the most accurate parameter for the prediction of severe cartilage degeneration. AUC comparison was performed based on pairwise comparison of ROC curves. MedCalc^®^ (Version 19.3, Medcalc Software Ltd, Ostend, Belgium) was used for statistical analysis. The significance level was set to *p* < 0.05.

Sample size calculation. The null hypothesis is that an AUC of 0.5 corresponds to random detection probability. The significance level and power of the test were set to 0.05 and 0.8, respectively. The estimated ratio between slight and severe degeneration was 3:1. To reach an AUC of 0.6 and higher (at least 0.1 higher than chance) a sample size of 352 patients was needed.

## 3. Results

### 3.1. Demographics and Average Measurements

Gender (f:m) and examination side (right:left) were equally distributed (Table 2). The average age and BMI of the study patients was 45.3 ± 22 years and 27.2 ± 5.4 kg/m^2^, respectively. The retropatellar compartment presented the highest level of cartilage degeneration with an average Outerbridge score of 1.9 ± 1.5, followed by the medial and lateral compartment with scores of 1.7 ± 1.5 and 1.5 ± 1.4, respectively. The average maximal knee diameter and cross-sectional area of the knee were 13.8 ± 1.4 cm and 131.6 ± 30.2 cm^2^, respectively. The examined parameters and corresponding results are summarized in Table 3.

### 3.2. Logistic Regression for Binary Variables

Gender, examined side and MRI scanner strength did not correlate with the evaluated outcome (severe degeneration = Outerbridge grades 3 and 4), proving therefore useless in predicting degeneration, with the following logistic regression formula:
Logit (severe degeneration) = −0.264 * M + 0.068 * R − 0.18 * T + 0.25

M: male patient = 1, female patient = 0; R: right knee = 1, left knee = 0; T: 1.5 T scanner = 1, 3 T scanner = 0. These variables demonstrated non-significant odd ratios for degeneration around 1 (Table 4).

### 3.3. Receiver Operating Characteristic Curves (ROC) for Non-Binary Variables

The parameter that allowed for the most accurate prediction of severe degeneration (Outerbridge grades 3 and 4) was the age, with an AUC of 0.865 ± 0.02 on the specificity-sensitivity plot. For an age over 41 years, sensitivity and specificity for severe degeneration were 82.8% (CI: 77.5–87.3%) and 76.4% (CI: 70.4–81.6%), respectively. The next best predictor for degeneration was the normalized cross section area of the knee (AUC = 0.767 ± 0.04), followed by BMI (AUC = 0.739 ± 0.02) and the normalized maximal diameter of the knee (AUC = 0.724 ± 0.05). The absolute diameters and cross-sectional areas showed AUCs in the range of 0.653–0.685. The Insall-Salvati index proved to be useless for making predictions. The AUC, sensitivity, specificity and relative criteria for all non-binary variables are listed in Table 5 and the ROC curves are depicted in Figure 4.

### 3.4. Comparisons of ROC Curves

Age has a significantly higher AUC for predicting severe degeneration than BMI (*p* < 0.0001), maximal diameter (*p* < 0.0001) and knee cross-sectional area (*p* < 0.0001). BMI showed a significantly better accuracy for detecting degeneration than maximal diameter (*p* = 0.0049) and knee cross-sectional area (*p* = 0.0008).

For the 122 patients that have been normalized by the maximal femur diameter (Figure 5), age did not demonstrate a significant difference in AUC compared to the normalized maximal diameter (*p* = 0.363) or the normalized knee area (*p* = 0.846). No combination of variables for a multicriteria ROC analysis reached a higher AUC than age alone.

## 4. Discussion

The purpose of our study was to assess the areas under the curve of multiple variables (age, height, weight, BMI, Insall-Salvati index, cross section area of the knee, vertical diameter, horizontal diameter, maximal diameter) as well as normalized cross-sectional area and normalized maximal diameter as biomarkers for the prediction of knee cartilage degeneration. The knee images used for this study were acquired using 1.5 T and 3 T MRI scanners. To grade cartilage changes, the modified Outerbridge classification was used. Both 1.5 T and 3.0 T scanners have been proven effective in detecting not only advanced cartilage loss but also mild cartilage changes [25], although previous studies have stressed the superiority of 3.0 T scanners for cartilage assessment [19,26].

Degenerative cartilage changes and subsequent loss in joint cartilage volume are among the first structural changes observed during OA progression, and they are always present in patients with advanced osteoarthritis [21]. 

Age and obesity are known risk factors for knee pain and osteoarthritis. Previous studies showed that degenerative knee changes increase with increasing obesity measured in terms of BMI and that cartilage loss extent is age-dependent and more pronounced in individuals aged 50+ years [9,19,27,28,29]. 

Our study results provide evidence for the fact that knee cartilage changes begin earlier in life with a cut-off for prediction of cartilage degeneration set to 41 years of age, making age the best surrogate biomarker for severe cartilage degeneration. 

Many age-dependent factors such as gradual strength loss of quadriceps muscle and increased oxidative stress in the knee joint microenvironment contribute to this process [30,31,32,33,34].

As for obesity, body weight excess promotes knee joint degeneration in a multifactorial way, including both mechanical and metabolic components. On the one side indeed, obesity constitute a chronic overload for the legs’ joints, and it also modifies the biomechanics and the axes load of the joints in the lower extremities. On the other side, metabolic factors are affected by obesity, e.g., the proliferation of chondrocytes, which is impaired in overweight individuals, or leptin secretion, a hormone regulating the food intake habits that was shown to also play a role in OA [10,35,36]. 

Previous attempts have been made to quantify the amount of adipose tissue surrounding the knee, beyond BMI, e.g., by measuring the anteroposterior and sinistro-dexter (medio-lateral) knee diameter as well as by measuring the surface of the knee cross-sectional area, and to evaluate these measurements as possible prognostic indicators for cartilage loss [8]. The rationale behind the evaluation of these parameters with respect to their ability to be prognostic markers was their potential use by clinicians as a simplified way to predict cartilage degeneration, performing the relevant measurements on the x-ray images, where the cartilage is not visible, thus avoiding a more expensive MRI examination, or even by measuring the diameters or the circumference of the knee directly on the patient’s body, making imaging unnecessary, at least at the moment a first risk assessment is indicated. Only a weak association was observed between the extent of cartilage loss, as graded by the Outerbridge scale, and the anteroposterior knee diameter, the sinistro-dexter diameter or the knee surface area.

In the present study, we included in our evaluation also the normalized cross-sectional area and the normalized maximal diameter of the knee by dividing the cross-sectional area and the maximal knee diameter by the maximal femoral diameter in a subgroup of 122 patients. The area under the curve was used to evaluate the different parameters as possible predictors. No combination of these parameters proved to be a better predictor of degeneration than age alone. In the 122 patients in which measurements were normalized to the maximal femur diameter, normalized diameter and area measurement showed the same ability to predict severe cartilage degeneration as the age variable did, whereas the absolute measurements of the diameter and area (*n* = 481) were significantly inferior as surrogate biomarkers than the variable age alone. This signifies that in order for the measurements to be as efficient as age in predicting cartilage degeneration, the measurements need to be normalized.

### 4.1. Strengths 

The large number of patients (481), the inclusion of all knee compartments in the cartilage assessment, the further analysis of knee cross-sectional area and maximal diameter upon normalization and the calculation of the ROC AUC values for all parameters are some of the strengths of our study, making the statistical results of the study more reliable and strengthening the conclusions.

### 4.2. Limitations

Axial images were not perpendicular to leg axis, which might have caused an overestimation of the diameters, circumference and knee cross-sectional areas. However, we counteracted the above-mentioned bias by simplifying the readout: the strictly vertical and horizontal diameters of the knee were measured and the knee cross-sectional surface was measured automatically by Osirix, allowing greater reproducibility. Furthermore, the Outerbridge classification is a radiological score and not a reliable direct measurement. Although the correlation between the Outerbridge score and arthroscopic or surgical knee findings is high [21], the clinical impact would still be the ground truth. Another limitation is that inter- and intra-reader variability assessments are missing due to the fact that no double reading was performed because of the large size of the dataset under investigation.

## 5. Conclusions

Our hypothesis that larger knee diameters and cross-sectional area but also gender, obesity (BMI), and the older age are correlated with a larger extent of knee cartilage degeneration proved correct. These parameters can therefore be used as surrogate biomarkers for knee cartilage loss in OA (Outerbridge grades 3 and 4), with age representing the most accurate biomarker, but not the only one. The measurement of the knee diameter directly on the patient’s body or on the patient’s knee x-ray images can substitute for the use of lengthier and more expensive imaging methodologies and constitute an easily accessible diagnostic tool that not only radiologists but also general practitioners or orthopedic surgeons can use, with a maximal knee diameter over 13 cm being an alarm bell for severe knee cartilage degeneration. 

## Figures and Tables

**Figure 1 medicina-59-00027-f001:**
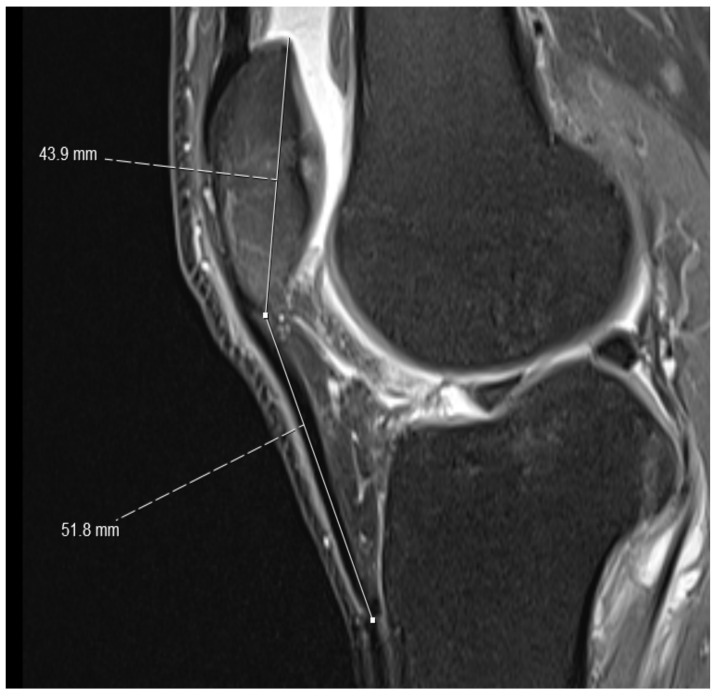
Normal Insall-Salvati index. For the knee in the figure, an Insall Salvati index of 1.18 was obtained by dividing the length of the patellar ligament (51.8 mm) by the patellar pole distance (43.9 mm).

**Figure 2 medicina-59-00027-f002:**
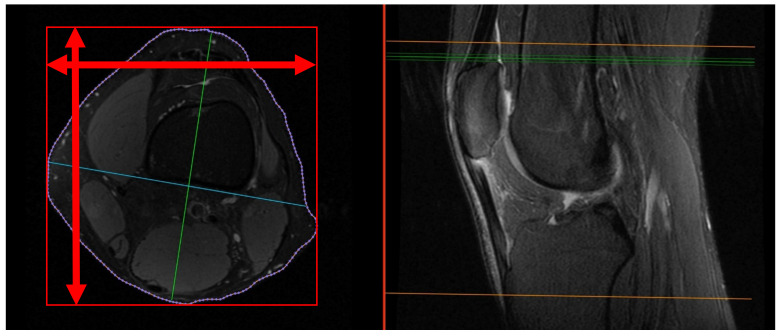
Knee diameters were measured on axial PD-weighted images at the exact level of the patella upper pole, indicated by the green line on the right image. Radiologists measured strictly antero-posterior = vertical and medio-lateral = horizontal (red arrows). The maximal diameter of the knee was not measured (green line on left image).

**Figure 3 medicina-59-00027-f003:**
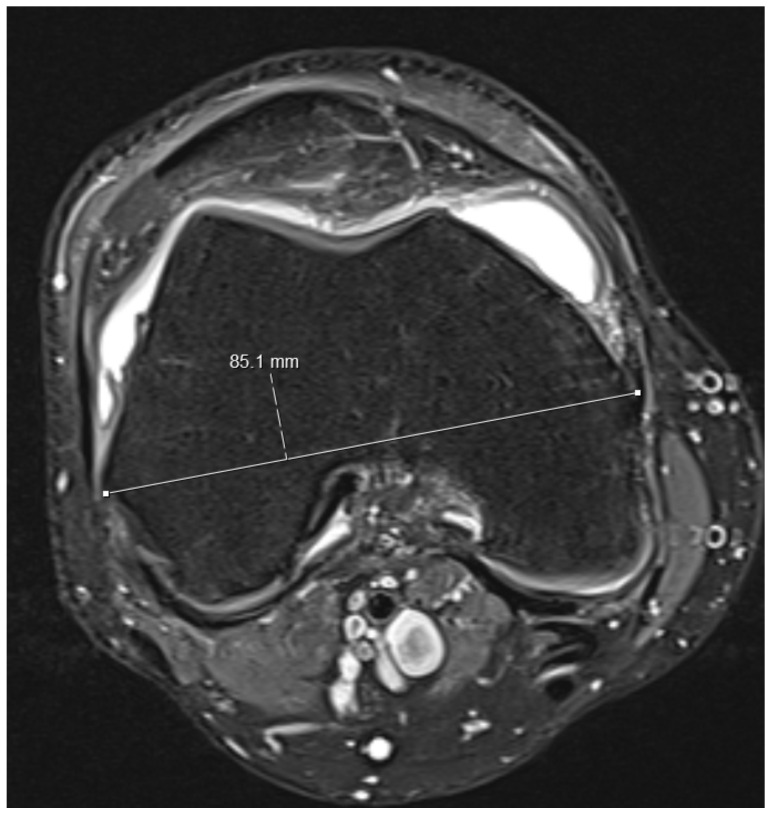
The largest axial femur diameter was measured at the level of the condyles as indicated in the figure.

**Figure 4 medicina-59-00027-f004:**
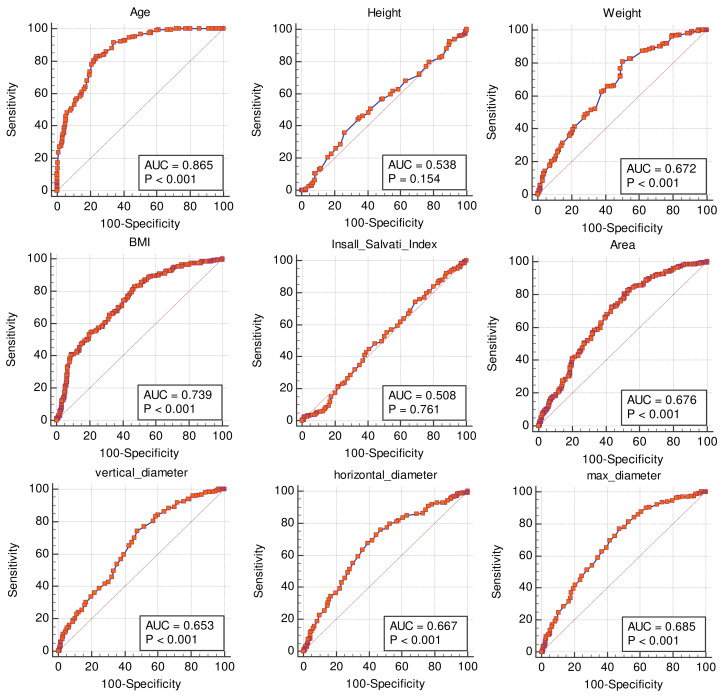
Areas under the curve of the non-binary variables. Top predictors for severe Outerbridge score before normalization are age, BMI and maximal knee diameter.

**Figure 5 medicina-59-00027-f005:**
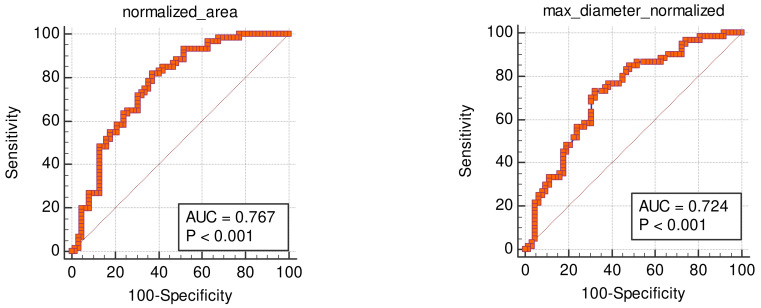
Normalizing the knee cross-sectional area and maximal knee diameter by dividing the measurements by the maximal femur diameter (at the level of the condyles) results in an increase in the respective AUCs.

**Table 1 medicina-59-00027-t001:** Modified Outerbridge classification.

Grade	Macroscopy	MRI
Grade 0	Normal cartilage	Normal cartilage
Grade 1	Rough surface; chondral softening, focal thickening	Inhomogeneous; high signal; surface intact; cartilage swelling
Grade 2	Irregular surface defects; <50% of cartilage thickness	Superficial ulceration, fissuring, fibrillation; <50% of cartilage thickness
Grade 3	Loss of >50% cartilage thickness	Ulceration fissuring, fibrillation; >50% of depth of cartilage
Grade 4	Cartilage loss	Full thickness chondral wear with exposure of subchondral bone

**Table 2 medicina-59-00027-t002:** Binary variables distribution.

Gender (f:m)	242:239
Knee (right:left)	241:240
MRI (1.5T:3T)	163:318

**Table 3 medicina-59-00027-t003:** Summary of non-binary variable statistics.

	N	Mean	Median	SD	25–75 P
**age (years)**	481	45.33	45	21.66	27 to 62
**BMI (kg/m^2^)**	481	27.21	27	5.39	24 to 30
**weight (kg)**	481	79.32	80	17.78	69 to 90
**height (m)**	481	1.70	1.70	0.11	1.63 to 1.78
**knee cross-sectional area (cm^2^)**	481	**131.63**	128	30.21	111 to 146
**horizontal knee diameter (cm)**	481	**13.42**	13.5	1.64	12.4 to 14.4
**vertical knee diameter (cm)**	481	**13.25**	13.1	1.46	12.3 to 14.1
**maximal knee diameter (cm)**	481	**13.78**	13.7	1.43	12.8 to 14.7
**Insall Salvati index**	481	1.08	1.08	0.16	0.98 to 1.18
**maximal Outerbridge grade per knee**	481	**2.28**	3	1.55	1 to 4
**maximal lateral Outerbridge grade**	481	**1.52**	2	1.41	0 to 3
**maximal medial Outerbridge grade**	481	**1.70**	2	1.54	0 to 3
**maximal retropatellar Outerbridge grade**	481	**1.89**	2	1.46	0 to 3
**maximal femur diameter (cm)**	122	8.29	8.4	0.71	7.7 to 8.9
**normalized knee cross-sectional area**	122	15.47	14.36	4.29	13.0 to 16.5
**normalized maximal knee diameter**	122	0.89	0.894	0.12	0.82 to 0.97

N: number of cases; SD: Standard Deviation; P: Percentile.

**Table 4 medicina-59-00027-t004:** Logistic regression of dichotomous variables for severe knee degeneration (Outerbridge > 2).

Variable	Odds Ratio	95% CI	Coefficient	Std Error	*p*-Value
Gender = m	0.768	0.53 to 1.10	−0.26	0.18	0.152
Knee side = right	1.071	0.75 to 1.54	0.07	0.18	0.710
MRI = 1.5 T	0.833	0.57 to 1.22	−0.18	0.19	0.348
Constant			0.25	0.21	0.241

CI: Confidence Interval, m: male; MRI: Magnetic Resonance Imaging; T: Tesla.

**Table 5 medicina-59-00027-t005:** Accuracy (AUC) of demographic and radiologic parameters for the prediction of severe knee degeneration (Outerbridge > 2).

Variable	AUC	Std. Error	*p*-Value	Criterion	Sensitivity	95% CI	Specificity	95% CI
age	0.865	0.02	**<0.0001**	>41 years	82.8	77.5–87.3	76.4	70.4–81.6
Height (m)	0.538	0.03	0.1535	≤1.66 m	45.1	38.7–51.6	65.0	58.5–71.0
weight	0.672	0.02	**<0.0001**	>72 Kg	80.7	75.2–85.5	49.8	43.3–56.3
BMI	0.739	0.02	**<0.0001**	>24.9	82.8	77.5–87.3	52.7	46.2–59.2
Insall-Salvati index	0.508	0.03	0.7613	>1.25	9.4	6.1–13.8	83.5	78.2–88.0
AREA (cm^2^)	0.676	0.02	**<0.0001**	>117.2 cm^2^	80.3	74.8–85.1	49.0	42.4–55.5
vertical diameter (cm)	0.653	0.02	**<0.0001**	>12.7 cm	74.2	68.2–79.6	52.3	45.8–58.8
horizontal diameter (cm)	0.667	0.02	**<0.0001**	>13.1 cm	73.0	66.9–78.4	55.7	49.1–62.1
max diameter	0.685	0.02	**<0.0001**	>13.3	77.1	71.3–82.2	52.3	45.8–58.8
max diameter normalized	0.724	0.05	**<0.0001**	>1.60	73.3	60.3–83.9	67.7	54.7–79.1
normalized area	0.767	0.04	**<0.0001**	>14.08	81.7	69.6–90.5	62.9	49.7–74.8

AUC: Area under the curve; CI: Confidence Interval; BMI: Body Mass Index.

## Data Availability

Data are available upon special request.

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
