# Peer review of "Knee Diameter and Cross-Sectional Area as Biomarkers for Cartilage Knee Degeneration on Magnetic Resonance Images"

_medicina, 2022, doi:10.3390/medicina59010027_

Round 1

Reviewer 1 Report

interesting idea but unfortunately this study has several major methodological flaws
(no reliability measures, non-homogeneous data, no sample size, incorrect statistical analysis)
scientific writing is somehow naive and should be deeply revised.

COMMENTS
-please be sure that the manuscript is revised by a proficient English speaker researcher
the language seems somehow naive
-please be sure that references are correctly formatted, relevant, and updated
-please be sure that the abstract is correctly formatted according to journal standards
-introduction is not focused on the topic and confusional
please stay focused on your study and explain what is known on the topic and why this study is needed
-please report the aims and hypothesis at the end of the introduction
-as an example of poor scientific writing, "adipose person" does not sound right, use BMI instead
the whole manuscript should be revised accordingly for similar mistakes
-line 62-63, this is exclusion criteria in the method section
-please clearly report the study design
-no reliability measures is an important limitation of this study
-please report the sample size calculation
-line 143-144: why the affected side and MRI magnetic field were put in the logistic regression
how could possibly these variables influence knee degeneration?
ask for help from a proficient statistician if necessary

Author Response

Thank you for your valuable input. We changed the manuscript accordingly. The study improved strongly. Please find the point-by-point answers in the attached doc.

Reviewer 2 Report

Dear Authors,

I read with interest your paper, as the incidence of knee osteoarthritis is increasing worldwide and therefore, strategies aiming to identify the disease in the early stages are important. However, there are some aspects which I consider that need improvement in your manuscript:

Introduction:

- please provide more information and references. The text is not fluent as you move very quickly for anatomic details to causes of depression. I would appreciate inserting more often single references, instead of groups after long paragraphs. 

- criteria of Outerbridge scoring system should not be included here. Also, I saw that the same information is repeated in Table 1. 

-lines 57-58 are unclear, words seem to be missing

- lines 62-63 should belong to Material and Methods.

Material and Methods:

- please describe how to calculated the necessary sample size

- I did not understand why you did not calculate intra- and interobserver reliability coefficients. This diminishes the quality of your results.

Results:

- the numbering of the figures is incorrect (there should be Figure 4 and 5, not 1 and 3).

Discussion: 

- for lines 227-229, please provide references.

References 4, 17, 20 are studies conducted on animal models, how do you compare them with studies on humans? Maybe you could find more suitable references. 

Author Response

Thank you for your valuable input. We changed the manuscript accordingly, which helped strongly to improve the paper .

Round 2

Reviewer 1 Report

thanks for your well performed revision of the paper

well done

Reviewer 2 Report

Dear Authors,

I appreciate your effort in improving the manuscript.